# Do Targeted User Fee Exemptions Reach the Ultra-Poor and Increase their Healthcare Utilisation? A Panel Study from Burkina Faso

**DOI:** 10.3390/ijerph17186543

**Published:** 2020-09-08

**Authors:** Yvonne Beaugé, Manuela De Allegri, Samiratou Ouédraogo, Emmanuel Bonnet, Naasegnibe Kuunibe, Valéry Ridde

**Affiliations:** 1Heidelberg Institute for Global Health, Medical Faculty and University Hospital, Heidelberg University, Im Neuenheimer Feld 365, 69120 Heidelberg, Germany; Manuela.DeAllegri@uni-Heidelberg.de (M.D.A.); nkuunibe@uds.edu.gh (N.K.); 2The Canadian Institutes of Health Research (CIHR), Ottawa, ON K1A 0W9, Canada; samioued@yahoo.fr; 3National Public Health Institute of Quebec (INSPQ), Quebec City, QC G1V 5B3, Canada; 4Department of Epidemiology, Biostatistics and Occupational Health (EBOH), Faculty of Medicine, McGill University, Montreal, QC H3A 1A2, Canada; 5French Institute for Research on Sustainable Development (IRD), Unité Mixte Internationale (UMI) Résiliences, 93143 Bondy, France; emmanuel.bonnet@ird.fr; 6Department of Economics and Entrepreneurship Development Studies, Faculty of Integrated Development Studies, University for Development Studies, P. O. Box 520, Wa, Upper West Region, Ghana; 7French Institute for Research on sustainable Development (IRD), Centre Population et Développement (CEPED), Universités de Paris, ERL INSERM SAGESUD, 75006 Paris, France; valery.ridde@ird.fr

**Keywords:** user fee exemptions, targeting, health service utilisation, performance-based financing, Burkina Faso

## Abstract

*Background:* A component of the performance-based financing intervention implemented in Burkina Faso was to provide free access to healthcare via the distribution of user fee exemption cards to previously identified ultra-poor. This study examines the factors that led to the receipt of user fee exemption cards, and the effect of card possession on the utilisation of healthcare services. *Methods:* A panel data set of 1652 randomly selected ultra-poor individuals was used. Logistic regression was applied on the end line data to identify factors associated with the receipt of user fee exemption cards. Random-effects modelling was applied to the panel data to determine the effect of the card possession on healthcare service utilisation among those who reported an illness six months before the surveys. *Results:* Out of the ultra-poor surveyed in 2017, 75.51% received exemption cards. Basic literacy (*p* = 0.03), living within 5 km from a healthcare centre (*p* = 0.02) and being resident in Diébougou or Gourcy (*p* = 0.00) were positively associated with card possession. Card possession did not increase health service utilisation (β = −0.07; 95% CI = −0.45; 0.32; *p* = 0.73). *Conclusion:* A better intervention design and implementation is required. Complementing demand-side strategies could guide the ultra-poor in overcoming all barriers to healthcare access.

## 1. Introduction

Despite Burkina Faso’s progress towards achieving Sustainable Development Goal 3 (SDG3), which seeks to ensure healthy lives and promote well-being for all at all ages, many people, especially the ultra-poor, still lack access to basic healthcare services due to the existence of user fees. According to the Ministry of Social Action and National Solidarity, ultra-poor people are defined as persons who are without sufficient financial and social means on a sustained basis and who are not able to take care of themselves [1].

User fees to pay for consultations, medications and laboratory work are still predominant at all levels of care in Burkina Faso [2]. According to the most recent Demographic and Health Survey, user fees are the most important barrier to healthcare access [3]. Health service utilisation is very low across the country, standing at 0.6 outpatient consultations per inhabitant per year [4]. Out-of-pocket expenditure in 2017 was estimated at 32% of the current total health expenditure [5].

In this context, targeted user fee exemptions for the ultra-poor (indigent) population is a policy instrument that is gaining prominence. Exemptions are intended to facilitate access to healthcare services by removing financial barriers at the point of service use [6]. Despite the growing implementation of user fee exemptions, little scientific attention is being paid to the ultra-poor in Burkina Faso and on the question of whether the utilisation of healthcare services effectively increases after targeted user fees exemptions have been implemented. Most evidence originates from other countries such as Cambodia, Cameroon, Morocco and Zambia which generally show an increase in service utilisation among the ultra-poor [7,8,9,10,11] or addresses other population groups [12,13,14]. In Burkina Faso, only Atchessi et al. (2016) conducted a pre–post study in one district and reported that the distribution of user fee exemption cards did not increase healthcare utilisation [15]. Reasons for the paucity of evidence include the extremely poor living conditions and the accompanied difficulties in recruiting the ultra-poor for research. However, what is known in terms of the rural ultra-poor is the generally low level of healthcare utilisation [16].

Given the very low healthcare utilisation among the ultra-poor, the government of Burkina Faso adapted various policies to explicitly exempt them from paying for healthcare services at the primary level; first in the context of the Bamako Initiative but also in consecutive years [17,18,19,20]. The current law on Universal Health Insurance (*loi sur l’assurance maladie universelle*) adopted in September 2015, also recognised the liability of the state to fully pay the user fees of the ultra-poor. However, nearly all policies specifically targeting the ultra-poor have so far been rather ineffective, due to chaotic implementation, technocratic processes and insufficient funding [17]. 

In addition to the many policy attempts, the Burkinabè government implemented several exemption pilot programmes on a smaller scale [21,22]. In 2014, Burkina Faso tested performance-based financing (PBF) [18,19] in combination with targeted user fee exemptions [20] on a larger scale in eight health districts (Diébougou, Batié, Kongoussi, Kaya, Ouargaye, Tenkodogo, Gourcy, Ouahigouya). Community-based targeting (CBT) was used to identify up to 20% of all individuals residing in the district as extremely poor. Community selection committees (CSC) were set up across the districts at the village level to select the poor. All identified ultra-poor were meant to receive an exemption card that proved their indigent status (Table 1) and allowed them to access all services included in the PBF benefit package free of charge [23]. However, evidence from the process evaluation indicated that not all initially selected ultra-poor had received an exemption card. The PBF benefit package covered maternal care services, general curative consultations, HIV services, tuberculosis services and family planning [21]. The healthcare facilities received a lump-sum to compensate for the loss of income from user fees (for details see Appendix A).

The objective of this study was to examine factors associated with the receipt of user fee exemption cards and understand which individuals were more or less likely to receive a card. Secondly, the study assessed whether the targeted user fee exemptions for the ultra-poor increased the health service utilisation, when the ultra-poor moved from no exemption card possession during the first round of the survey in 2015 to exemption card possession in the second round of the survey in 2017. Based on previous evidence on the effects of user fee reduction/removal policies on use of curative [22], delivery [12] and child care services [13] in Burkina Faso, the underlying hypothesis was that card possession would result in an increase in service use for exempted ultra-poor. Anderson’s behavioral model on health service use was applied [24].

## 2. Materials and Methods 

### 2.1. Study Design and Population

This study relied on a pre–post panel design using a dataset of 1652 ultra-poor surveyed in the first round between February and April 2015 and in the second round between February and March 2017. The respondents surveyed were resident in Diebougou, Gourcy, Kaya and Ouargaye health district in the south-west, centre-north, centre-east and northern region of Burkina Faso. About 45% of the population lives below the national poverty line of USD 1.90 a day [25]. The northern region has the highest poverty incidence and accounts for 70.4% compared with 9.6% in the central region [26]. The surveys were conducted during the dry season when it was more likely that the ultra-poor would be available for interviews than during the agricultural season. All of the respondents were initially identified as ultra-poor by the community-based targeting process embedded within the PBF-programme. 

The respondents were selected using a multistage random sampling technique. The first stage involved the random selection of four out of eight PBF districts with CBT. These four districts comprised a total of 1,032,541 inhabitants, of which 51,267 people were identified as the ultra-poor by the CBT. The second stage was the random selection of communes and villages within each district. Villages were only included if they contained a minimum of 10 ultra-poor identified by the CBT. Fifty-eight villages met this criterion. The third stage involved the selection of ultra-poor aged 18 and above and whose name was on the original ultra-poor list and were recruited for the survey. 

Ultra-poor were excluded if they could not give informed consent or were unable to understand or answer survey questions. Further details on sampling procedures are described in two previous manuscripts [27,28], which assessed the characterisation of the rural ultra-poor population and their mental health needs. 

The data collection tool was a structured closed-ended questionnaire which assessed the ultra-poor’s sociodemographic characteristics, their health and health service use, and also acquired information on their mental health and cognitive functioning.

Trained enumerators went to the place of residence of the ultra-poor individuals and asked for their verbal consent to administer the questionnaire face-to-face in the local language. Tablets with an Open Data Kit (ODK) software were used for the administration of the questionnaire, whereby the entered data was transferred to the central database daily [27,28]. The interview duration was on average one hour. Ethical clearance was granted by the Comité National d’Éthique pour la Recherche en Santé (CNERS) in Burkina Faso (Decision No. 2019-01-004). In the following text, the ultra-poor are called respondents.

### 2.2. Variables and their Measurement

Table 2 provides an overview of all variables included in the analysis, their measurement and the hypothesised direction of the coefficient for both Model 1 and 2. Variables were selected based on the availability in the dataset and in accordance with the Anderson behavioral model [24], which has been widely used to explain the determinants of health service utilisation. According to the model, factors can be divided into three groups as predisposing, enabling and need-related. 

Predisposing factors were age, sex and marital status. Enabling factors were educational level, basic literacy and distance to the health facility. In model 2, the main explanatory variable ‘possession of user fee exemption card’ was added as an enabling factor. Need factors were self-rated perceived health and disability. 

Predisposing factors: Age (in years) and household size (number of members) was a continuous variable. Sex was a dichotomous variable (male/female). Marital status was a categorical variable with five categories (single, monogamous married, married polygamous, widowed, divorced/separated). The original variable was transformed into a binary variable (All else and married) for multivariate analyses in order to expose the particular vulnerability associated with an unmarried status. Status in the household was a categorical variable with 11 categories (Household head; spouse; brother/sister; son/daughter; nephew/niece; Grandson/daughter; father/mother; cousin; son/daughter in law; mother/father-in-law; other parent; other link). This variable was dichotomised to express the superiority associated with being a household head. 

Enabling factors: Educational level was a categorical variable with 16 categories (1 none; 2 nursery school; 3 CP1 4 CP2; 5 CE1 6 CE2; 7 CM1; 8 CM2; 9 Sixième; 10 Cinquième; 11 Quatrième; 12 Troisième 13 Seconde; 14 Première; 15 Terminale; 16 Supérieur). As done by previous studies performed in a rural African context [29], the original variable educational level was transformed into a binary variable (no education and education). Less than 6% of the study samples (ultra-poor population) received any form of education. The category ’Education’ contained all those respondents who attained higher education than a nursery school. Basic literacy was defined as the ability to write and was a dichotomous variable (Yes/No). The variable distance was dichotomised to reflect the standard of having a primary health facility within and outside a radius of 5 km as set by the World Health Organisation. For Model 2, the possession of user fee exemption card was added as a dichotomous variable (Yes/No). If respondents had been identified as ultra-poor but did not receive their exemption card due to a default in the system, they could not prove their ultra-poor status and were thus theoretically not eligible for free services. They were coded as respondents without an exemption card (exemption card – 0). This information was self-reported.

Need factors: Self-rated perceived health was a categorical variable (good, medium, bad). The variable was transformed into a binary one (All else/Good). Disability was a dichotomous variable (Yes/No). 

Additionally, the district was added as another explanatory variable because of slight variations in the implementation of the targeting and exemption mechanism across the districts (e.g., transportation of exemption cards to respective districts – different time points, see Table 1) that could have impacted the utilisation of healthcare services by the poor (1 = Kaya; 2 = Ouargaye; 3 = Diebougou; 4 = Gourcy). Time dummies (0 = 2015: 1 = 2017) were created to control for time variations of the dependent variable across the panels. Information about the ultra-poor’s consumption or income/assets (study population = ultra-poor without financial means) was not available in the dataset. The expected directions of the coefficients (Table 2) were informed by previous evidence on the determinants of healthcare utilisation among rural and vulnerable populations [15,30,31,32]. 

### 2.3. Statistical Analysis

The analysis was operationalised using the statistical package STATA version 15.0 (Stata Corp, Lakeway Drive, College Station, TX, USA). First, descriptive and comparative analysis was performed to determine the characteristics of each study sample at baseline (2015) and endline (2017) separately. The Chi-square, the t-test and the two-sample Kolmogorov-Smirnov test were used to determine whether the baseline and follow-up sample had the same statistical distributions. The significance level was set at α ≤ 0.05.

Second, two distinct models were used to: a) assess factors that influence the possession of user fees exemption card (Model 1); and b) determine the effect of this card possession on healthcare service utilisation among those who reported an illness six months before the surveys (Model 2). The choice of working with two separate models, (accounting in the second model for all possible observable confounders identified in the first one), resulted from the fact that the dataset did not provide a valid instrument for the application of an effective two-part joint model [33,34]. Having defined a binary outcome variable for both models (Yes/No), a multiple logistic regression was fitted for Model 1 and a regression analysis using a random-effects model for panel data using two time periods (2015 and 2017) for Model 2. 

The outcome variable for model 1 was defined as ‘the possession of user fees exemption card’ based on the survey question “Have you received a card about a year ago that you can present at the healthcare centre to receive care for free? - Yes or No”.

The outcome variable for model 2 was defined as ‘utilisation of healthcare services’ for those who reported an illness in the last six months. This variable referred to whether the respondent went to the healthcare centre six months before the survey irrespective of what kind of services were used (in-patient or outpatient)—Yes or No. Healthcare centre refers to either primary healthcare centres (CSPS) or district level hospitals. The targeted exemptions cards were earmarked only for formal healthcare services provided by the CSPS or district level hospitals. 

For Model 1, a multiple logistic regression analysis was performed using only endline data (2017) (during the first round of the survey in 2015, nobody could have received an exemption card yet). The regression equation can be found in the Appendix A.

For Model 2, a regression analysis was performed using a random-effects model [35,36] clustered at the individual level and restricted to individuals reporting an illness episode in the preceding six months using baseline (2015) and endline data (2017). The regression equation with more detailed information on the choice of the model can be found in Appendix A. 

Both models were estimated using the same set of explanatory variables outlined below. In Model 2, the possession of exemption card was included as the main explanatory variable, used as a proxy for being entitled to free healthcare services at the healthcare centre. The coefficients were estimated with a 95% CI (Confidence Interval). 

Additionally, the respondents were geolocated. The information was integrated into the Geographic Information System’ ESRI ArcGis’ version 10.6 (ESRI Inc., Redlands, CA, USA) to better understand patterns in the distribution of the exemption cards. The Euclidean distances were estimated between each respondent and the nearest healthcare centre, using direct lines [37]. The point analysis (location of ultra-poor) and the kernel density estimator method was applied [38]. The densities represent the concentration of selected ultra-poor within a radius of 2000 m.

## 3. Results

Of the 1652 (100%) respondents recruited for the baseline survey, 1260 (76.27%) completed the follow-up survey in 2017. In 2017, 124 (32%) respondents were lost to follow up, and 144 (37%) were physically absent on repeated visits. Respondents who were unable to take part in the follow-up survey were excluded from the study: 10 (3%) suffered from an illness; 5 (1%) were at an advanced age; 8 (2%) were mentally sick; 6 (1%) had an auditory handicap; 90 (23%) were deceased, and 5 (1%) refused to respond to the questionnaire The number of observations available for the analysis was 1652 in 2015 and 1260 in 2017, resulting in an unbalanced panel data set of 2912 observations.

### 3.1. Characteristics of the Study Sample

Table 3 reports the descriptive and comparative statistics for all variables included in the analyses for 2015 and 2017. At baseline, the mean age of the sample was 55.13 years (SD = 16.96), with 67.6% being females. The majority of the respondents lived in Diébougou district (33.17%), were not literate (93.70%), had no education (94.79%); and indicated that they were not suffering from any form of disability (76.45%). About 60% were married, 42.80% were head of the household and 75.85% lived within the 5 km radius to the nearest healthcare centre. There was a high geographical concentration of the selected respondents around the area of a primary health facility (CSPS) in Diébougou, Gourcy, Kaya and Ouargaye (Figure 1, Figure 2, Figure 3 and Figure 4). There was generally a lower geographical concentration the further the respondents lived from a CSPS. At least 75.51% of the respondents reported the receipt (card possession) of the user fee exemption card during the follow-up survey. Comparing the reported frequencies of 2015 with those of 2017, there was a significant difference for the variables age (*p* = 0.00), household size (*p* = 0.00), perceived health (*p* = 0.01), illness-reporting (*p* = 0.00) and utilisation of healthcare services (*p* = 0.05). The average household size in 2015 was 1.61 (SD = 1.58) members compared to 2.57 (SD = 1.97) in 2017. A total of 19.49% of the respondents in 2015 reported being in good health compared with 23.49% of respondents in 2017 (*p* = 0.01). In 2015, 70.70% of the respondents reported at least one illness episode in the last six months, compared with 62.78% of respondents in 2017 (*p* = 0.00). At baseline, 64.21% reported the utilisation of healthcare services at the healthcare centre, compared with 59.92% at follow-up (*p* = 0.05). 

### 3.2. Results from the Regression Model on User Fee Exemption Card Possession (Model 1)

Table 4 presents the results of the model identifying the determinants of user fee exemption card possession (Model 1). Basic literacy (*p* = 0.03), distance below 5 km to the nearest healthcare centre (*p* = 0.02) and the residency in the health district Diébougou (*p* = 0.00) and Gourcy (*p* = 0.01) were positively associated with card possession. Age, sex, marital status, status in the household, household size, education, perceived health and disability were not significant determinants of card possession. 

### 3.3. Regression Model on Service Use (Conditional upon Reporting Ill) (Model 2)

Table 5 presents the results of the random-effects model predicting utilisation of healthcare services conditional upon illness reporting in relation to possession of user fee exemption card while controlling for all other explanatory variables (Model 2). No association was found between possession of user fee exemption card and the utilisation of healthcare services (*p* = 0.73). In addition, education, basic literacy, marital status and distance were also not associated with the utilisation of healthcare services. Being the household head (*p* = 0.00), being male (*p* = 0.04), and a greater household size (*p* = 0.02) were positively associated with utilising healthcare services, while better perceived health was negatively associated. In contrast, having a disability (*p* = 0.00) and being advanced in age (*p* = 0.00) was negatively associated with utilising healthcare services. 

## 4. Discussion

This study makes an important contribution to the existing evidence by using an extensive panel data set of ultra-poor respondents (N = 1260). These respondents were monitored before and after the introduction of targeted user fee exemptions. The study examined which factors were associated with the receipt of user fee exemption cards and the effects of this card possession on their utilisation of healthcare services. Compared with the use of single cross-sectional designs, which usually suffer from nonequivalence between control and intervention groups, the study was able to draw a more precise estimation of effects. Bearing in mind the methodological advantages of the applied research design, this study offers valuable guidance to any governments and donors aiming at exempting the poorest from user fees.

A core finding is that the majority of the identified ultra-poor (75.51%) received the exemption cards whereby the possession of exemption cards was positively associated with basic literacy, distance below 5 km to the nearest healthcare centre and the residency in the health district Diébougou and Gourcy. Contrary to the original hypothesis, the findings indicated that the possession of the exemption cards did not increase their utilisation of healthcare services. Being the household head, being male, having bad perceived health, lower age, absence of a disability and a greater household size were positively associated with utilising health services.

The findings seem to contradict the conclusions drawn from previous studies performed in other settings that suggested a substantial increase in service use by the poor after the introduction of either user fee exemptions at the national level or targeted user fee exemptions implemented on a smaller-scale project basis [9,10,17,39,40,41]. For instance, a multilevel interrupted time series analysis of routine monthly utilisation statistics during 2006–2013 examined the impact of Cambodia’s Health equity fund on the utilisation of public health facilities and demonstrated an increase in the utilisation of primary and secondary care services by the poor [9]. However, the national scheme in Cambodia also addresses nonfinancial barriers and provides beneficiaries reimbursements for transportation costs to the healthcare facility or daily food allowances for caretakers [9] which has not been the case in Burkina Faso. Evidence from several west African countries on pilot fee-exemption interventions has also generally drawn positive results and demonstrated a rise in service utilisation among the poor [17,40,41,42]. A recent study by Cottin (2018) relied on a combination of propensity score matching with panel difference in differences (DID) and estimated a modest positive effect of a nationwide fee waiver programme on healthcare utilisation by the poor in Morocco [10]. However, none of these studies on the poor used a panel-level design to measure the effect of user fee exemptions. 

The findings are consistent with some studies assessing the effects of targeted user fee exemptions for the poor. For instance, using a pooled synthetic control method, Lepine et al. (2018) reported that the user fee removal in Zambia had not resulted in an increase in healthcare utilisation by the ultra-poor [43]. Compared with Cambodia’s land area of 181,035 km^2^, Zambia is four times bigger (land area: 752,618 km^2^), an important characteristic that might have contributed to the differences in the impact of the user fee removal across the countries, since the population (Cambodia: 15 million; Zambia 17 million) is spread over a larger area making access to healthcare services more difficult. Atchessi et al. (2014) conducted a pre–post study in Ouargaye (Burkina Faso) and reported an increase in health service utilisation among the ultra-poor from 2010 to 2011, which was, however, not associated with the distribution of exemption cards [15]. In line with our findings, the study also argues that sociocultural factors such as gender and cultural beliefs, as well as affordable transportation, might have been more influential determinants. 

### 4.1. The Role of Intervention Design and Implementation Failures

To better understand why the possession of user fees exemption cards did not increase the utilisation of healthcare services by the ultra-poor, the findings need to be interpreted in relation to the context of the intervention and its implementation. 

First, it is important to consider that implementers had to reduce the reimbursements price levels (including the financial incentives to reach out to the poor) for all services twice due to budgetary constraints [19]. Looking at the first 18 months of implementation (January 2014 to May 2016), Turcotte-Tremblay et al. (2017) reported that some healthcare providers were dissatisfied with the compensation received for treating the ultra-poor. They argued that since this population is affected by multiple morbidities, case-based lump-sum reimbursements set around the average cost of treatment were not sufficient to cover their actual health provision costs [19]. Therefore, it is hypothesised that provider perceived incentives were too small and providers were for this reason not motivated enough to take the initiative to attract the ultra-poor to the facilities as intended by the PBF programme. While further investigation is certainly needed, general adjustments of the reimbursement price levels are advisable, taking into account the complex morbidity-profile of the ultra-poor. Appendix A shows further information on the reimbursement procedure and Appendix A and Appendix A show the list of quantity indicators included in PBF design.

Additionally, due to significant delays in reimbursements, some healthcare facilities charged poor-patients irrespective of their exemption card [19]. At the same time, it needs to be noted that 25% of the initial identified ultra-poor have never received an exemption card, especially those living remotely from the health facility and those being less literate. It is thus not surprising that the intervention impact lags behind expectations. These circumstances suggest a need for adherence to implementation guidelines and a concentration of efforts to reach those remote from the healthcare centre. 

Another design element that might explain the reported lack of effectiveness of the intervention relates to the possibility of gaming/fraud by healthcare providers that can occur as an unintended consequence of PBF. This concern has led PBF designers to introduce a ceiling that rationed the services delivered to the ultra-poor in the targeted districts to a maximum of 10% of all consultations in health facilities [18,44,45].To understand this decision better, it is important to recall that the initial identification and targeting process allowed that up to 20% of the individuals in the health facility catchment area could be identified as ultra-poor and eligible for an exemption card. The community selection committees, however, only selected between 5% and 10%. Only a very high incidence of disease would lead the ultra-poor to account for more than 10% of all services provided. The imposition of the ceiling might have cautioned providers towards the provision of healthcare for the poor, resulting in the observed limited access rather than acting only as a deterrent to fraud and gaming, as originally expected. It is interesting to note that a parallel study looking specifically at misreporting suggests that contrary to expectations, extensive gaming and fraud are unlikely to have taken place in this setting. The study observed discrepancies in quantity reporting that were generally small and equally oriented towards under- and over-reporting [46].

### 4.2. Equity to Sccess to Healthcare is in the Eye of the Beholder

The fact that the study found no significant effect of the user fee exemption cards on the healthservice utilisation undoubtedly questions the design and content of the intervention, especially if one takes into account the financial and economic costs of identifying each ultra-poor beneficiary (USD 6 and USD 12 respectively) [21]. Furthermore, the user fee exemption was not a standalone project, but embedded within a broader PBF intervention that already complemented demand-side (user fee exemptions) by supply-side incentives (PBF) aimed at addressing inequalities in access to care in a more holistic way. It is especially against this background that these findings are alarming, although they echo results from previous studies that show that equity measures implemented alongside PBF fell short of reducing the equity gaps [47,48,49,50] with few exceptions [11,51]. The implications of these results for implementers and the government are that existing PBF strategies need to be better customised to fit the specific needs of the poor. User fee exemptions indeed represent a first step toward narrowing the equity gap. However, to receive the anticipated outcome and not waste resources, it is vital that future research explores and informs policymakers about the role and contribution of all relevant financial and nonfinancial barriers to healthcare access for the poor [52].

Interestingly enough, the findings suggested that in this specific context, it is not only the financial but the individual dispositions such as the position in the household, household size, perceived health status, age and the existence of a disability that might be more influential determinants of health service utilisation among the poor. This is in line with the theoretical models and frameworks that explain the complex nature of access to care and the multiple determinants of health service utilisation [24,53,54]. All of them stress that access to and the utilisation of healthcare services are dependent on not just the financial means of the poor. 

Despite the well-known complexity of the issue, policymakers and donors often tend to overemphasise the importance of financial access, as its degree of mutability is high opposed to, e.g., changing norms and social structures. Yet, equity to access to care is in the eye of the beholder [24], and it is ultimately the ultra-poor who can determine best what factors explain their utilisation. Hence, to promote equitable access to healthcare, global health actors and governments must take local contexts into account and adapt to these realities when designing public health interventions and, ultimately, policies [52,55]. To guide policy, future research with the application of mixed-method approaches, needs to focus on assessing the local perspective on the role and interrelation of various financial and nonfinancial barriers to access and utilisation comprehensively. 

In light of the results, one complementary strategy to the existent measures could be to better address gender inequalities through empowerment-based interventions, since women are still less likely to utilise healthcare services due to limited decision-making power [56]. To radically improve women’s capability to make health decisions, governments will have to go beyond mere reforms within the healthcare sector and introduce social and economic policies that strengthen women’s positions in society as a whole [57]. A barrier-focused intervention could introduce patient navigators within the primary healthcare system who serve as a link between the poor and healthcare provider by determining barriers to utilisation of services and coordinating and facilitating needed care [58]. This might be particularly effective not only for females but also the elderly; the study highlighted their decreased likelihood of utilising healthcare services. Another important factor in increasing utilisation rates is breaking down the transportation barrier, which remains a significant challenge for the ultra-poor.

## 5. Study Limitations

The findings should be interpreted in light of the study limitations. First, the study suffered from a high attrition rate, which resulted in a follow-up sample that became moderately biased towards having healthier participants and thus lower illness reporting and health service utilisation compared to its baseline counterpart. Sample attribution also entails the loss of a certain degree of statistical power. This attrition, however, appears inevitable given that prior research has indicated that the ultra-poor are more likely to be people of older age, people who experience severe illness or disability [27]. Second, due to too little variation in the main explanatory variable—Possession of user fee exemption card—the initial analytical approach had to be changed. The study applied a random instead of a fixed-effect model, an approach which would have allowed to control also for unobservable individual time-variant characteristics. However, having applied clustering at the individual level, the study obtained comparably accurate estimations. Third, the dataset might have been subject to illness reporting bias. As previously done by Schoeps et al. (2015) [59], the study controlled for all possible observable confounders to limit the extent to which working with the truncated sample of individuals having reported an illness episode might have affected the effect estimation.

Similarly, it would have been desirable to address the potential effect of endogeneity of possession of user fee exemption card on the estimates through the application of an effective two-part joint model. Still, due to the inability to identify in the dataset a valid instrument [33] it was not possible to do so. Furthermore, the study relied on self-reported information on illness and utilisation of healthcare services which are not 100% flawless. Lastly, data were collected retrospectively with a recall period of six months. Hence, the information on illness reporting and utilisation of healthcare services was subject to recall biases. 

## 6. Conclusions

The ultra-poor are the most vulnerable and underserved population in sub-Saharan Africa. By deciding to implement targeted user fee exemptions, Burkina Faso has taken a critical step to overcome barriers to equitable health service access. Against the original aim of the intervention, the study found that the utilisation of healthcare services by the ultra-poor was not responsive to the introduction of targeted user fees exemption. This finding, however, does not undermine the importance of such strategies to pursue UHC per se but implies that there are other more or equally important underlying barriers to universal healthcare access than financial ones, especially in settings where initial inequalities are large. Although the results are based on a small sample and relate to a limited geographic scope, they are highly relevant since the ultra-poor are a severely underrepresented group in the scientific landscape because of the difficulty in reaching them. Accordingly, the study offers valuable practical and political guidance which has long been overdue. The study ultimately serves the development of future public health interventions to truly leave no one behind, a principle at the heart of the 2030 Agenda for Sustainable Development. To improve the conditions for the poor effectively, three policy recommendations can be made. First, it is crucial to gain a precise local understanding of the relevant barriers of access to healthcare services for the ultra-poor. Second, it necessary to initiate dialogue with healthcare providers to find common ground on reimbursement price levels. Third, it is essential to prepare carefully, plan, and implement and fund user fee exemptions for the ultra-poor along with additional demand-side measures such as patient navigation to address all relevant barriers to healthcare access simultaneously. 

## Figures and Tables

**Figure 1 ijerph-17-06543-f001:**
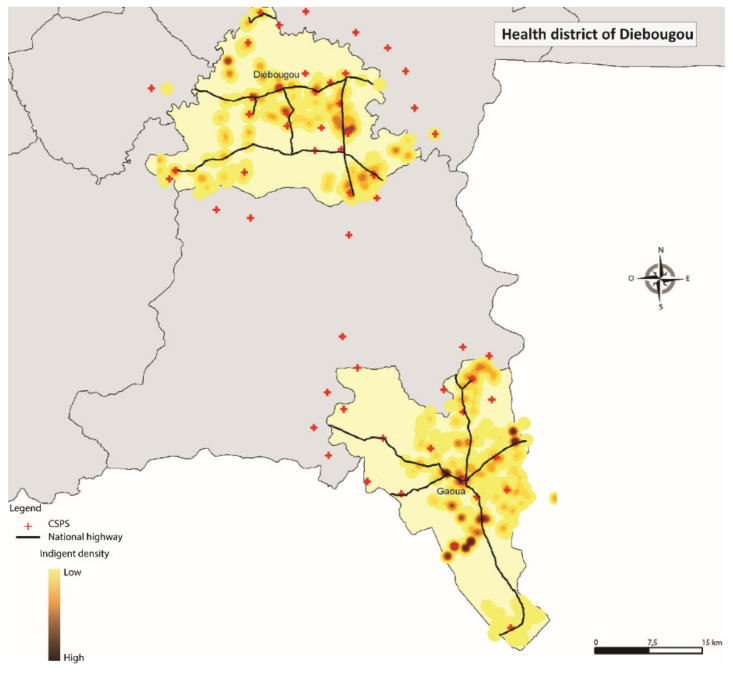
Geographical concentration of the respondents in Diébougou.

**Figure 2 ijerph-17-06543-f002:**
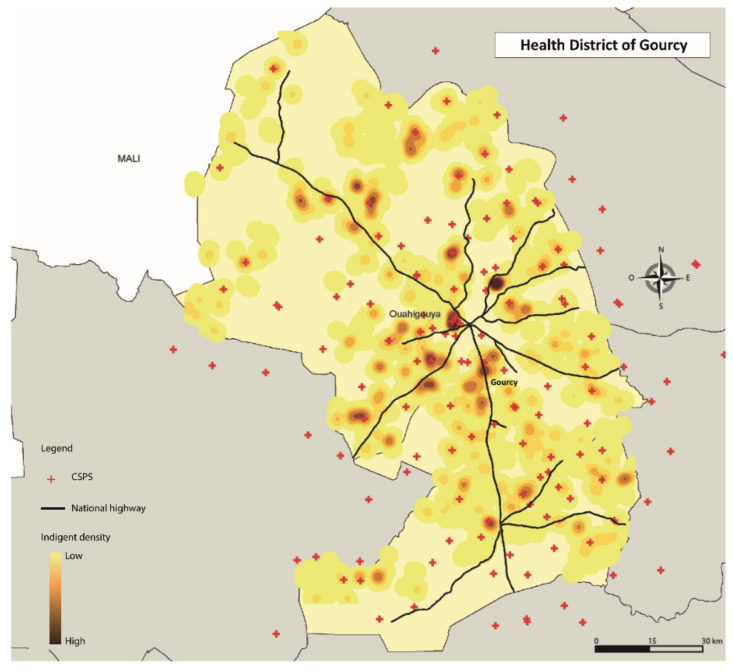
Geographical concentration of the respondents in Gourcy.

**Figure 3 ijerph-17-06543-f003:**
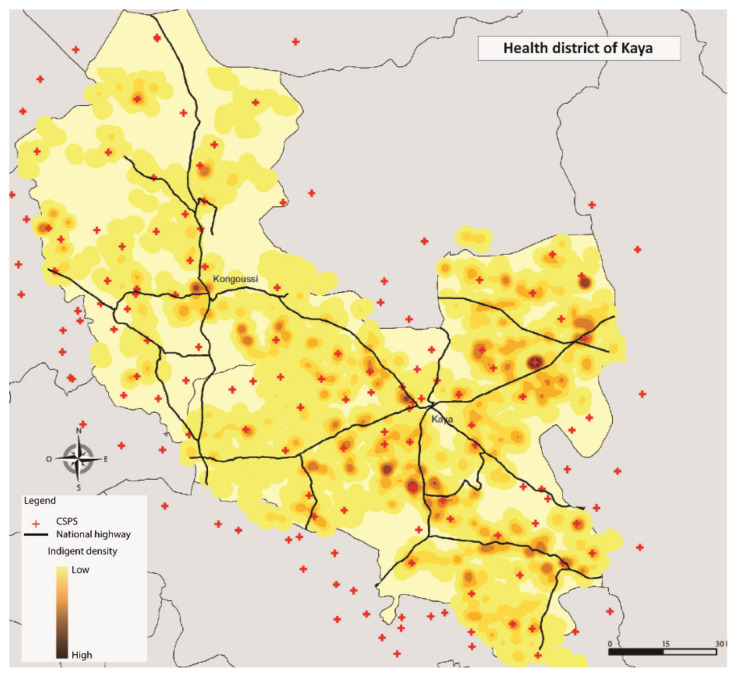
Geographical concentration of the respondents in Kaya.

**Figure 4 ijerph-17-06543-f004:**
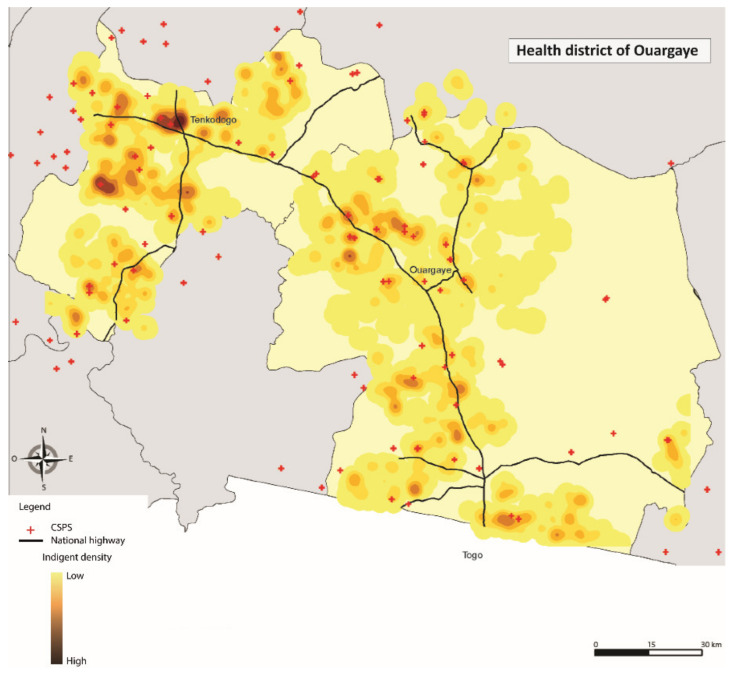
Geographical concentration of the respondents in Ouargaye.

**Table 1 ijerph-17-06543-t001:** Population, number and percentage of identified ultra-poor, and reception of exemption card by district.

District.	Population	Selected Ultra-Poor	Percentage of Selected Ultra-Poor (%)	Month Exemption Card Received by the District
**Diébougou**	69,062	6034	9	February 2016
**Gourcy**	132,280	5879	4	June 2016
**Kaya**	554,117	22,889	4	November 2015
**Ouargaye**	277,082	16,465	6	December 2015
**Tenkodogo**	216,190	18,769	9	December 2015
**Kongussi**	343,434	6076	2	November 2015
**Ouahigouya**	114,294	19,937	17	June 2016
**Batie**	39,330	6560	17	February 2016

**Table 2 ijerph-17-06543-t002:** Variables, their measurement and hypothesised direction of the coefficient for Model 1 and 2.

Variables	Measurement	Hypothesised Direction of the Coefficient Model 1	Hypothesised Direction of the Coefficient Model 2
**Outcome**			
Model 1: Possession of user fee exemption card	0 = No		
	1 = Yes		
Model 2: Utilisation of healthcare services	0 = No		
	1 = Yes		
**Predisposing factors**			
Age (continuous)	18–98 (years)	+	+
Sex	0 = Male		
	1 = Female		
Marital status	0 = All else	+	+
	1 = Married		
Status in the household	0 = No	+	+
	1 = Yes		
Household size (continuous)	1–12 (member)	+	+
**Enabling factors**			
Possession of user fee exemption card	0 = No	NA	+
	1 = Yes		
Education	0 = No	+	+
	1 = Yes		
Basic literacy	0 = No	+	+
	1 = Yes		
Distance to the nearest healthcare centre	0 ≤ 5 km	-	-
	1 ≥ 5 km		
**Need factors**			
Health status	0 = All else	+-	-
	1 = Good		
Disability	0 = No	+-	+-
	1 = Yes		
Health district	1 = Kaya	+-	+-
	2 = Ouargaye		
	3 = Diebougou		
	4 = Gourcy		
Time	0 = 2015	+	+
	1 = 2017		

NA indicates not applicable. + indicates a positive association is expected. - indicates a negative association is expected.

**Table 3 ijerph-17-06543-t003:** Comparison of the study sample characteristics 2015 and 2017.

Variables		2015 (N = 1652)		2017 (N = 1260)		Chi2 and *t*-Test	KS-Test	
***Outcome***		Frequencies	%	Frequencies	%	*p*-value	D-value	*p*-value
**Illness reporting**								
	No	484	29.30	469	37.22	0.00	0.08	0.00
	Yes	1168	70.70	791	62.78			
**Health service utilisation**								
	No	418	35.79	317	40.08	0.05	0.08	0.00
	Yes	750	64.21	474	59.92			
***Predisposing factors***								
**Age**		55.13 (mean)	16.96 (SD)	57.22 (mean)	16.95 (SD)	0.00 (t-test)	0.10	0.00
**Gender**								
	Male	535	32.38	403	31.98	0.82	0.00	1.00
	Female	1117	67.62	857	68.02			
**Marital Status**								
	All else	678	41.04	491	38.97	0.26	0.02	0.92
	Married	974	58.96	769	61.03			
**Household head**								
	No	945	57.20	711	56.43	0.68	0.01	1.00
	Yes	707	42.80	549	43.57			
**Household size**		1.61 (mean)	1.58 (SD)	2.47 (mean)	1.97 (SD)	0.00 (t-test)	0.20	0.00
***Enabling factors***								
**Exemption card possession**								
	No	1652	100.00	306	24.29	NA ^1^	0.76	0.00
	Yes	0	0.00	954	75.51			
**Education**								
	No	1566	94.79	1187	94.21	0.49	0.01	1.00
	Yes	86	5.21	73	5.79			
**Basic literacy**								
	No	1548	93.70	1187	94.21	0.57	0.01	1.00
	Yes	104	6.30	73	5.79			
**Distance to the nearest healthcare centre**								
	< 5 km	1253	75.85	940	74.60	0.44	0.01	1.00
	> 5 km	399	24.15	320	25.40			
***Need factors***								
**Health status**								
	All else	1330	80.51	964	76.51	0.01	0.04	0.20
	Good	322	19.49	296	23.49			
**Disability**								
	No	1263	76.45	992	78.73	0.15	0.02	0.85
	Yes	389	23.55	268	21.27			
***Additional variables***								
**Health District**								
	Kaya (1)	400	24.21	283	22.46	0.41	0.12	0.98
	Ouargaye (2)	423	25.61	354	28.10			
	Diebougou(3)	548	33.17	412	32.70			
	Gourcy (4)	281	17.01	211	16.75			
**Time**								
	2015	1652	100.00	0	0.00	NA	NA	NA
	2017	0	0.00	1260	100.00			

^1^ NA indicates not applicable.

**Table 4 ijerph-17-06543-t004:** Regression model on user fee exemption card possession.

Variable	Regression Coefficient (β)	Std Error	*p*-Value	[95% CI]
***Predisposing factors***				
Age	0.00	0.00	0.98	−0.01 0.01
Sex	−0.19	0.19	0.31	−0.56 0.18
Marital status	−0.07	0.17	0.69	−0.39 0.26
Status in the household	−0.23	0.17	0.18	−0.57 0.10
Household size	0.06	0.04	0.11	−0.01 0.14
***Enabling factors***				
Education	−0.14	0.38	0.72	−0.88 0.61
Basic literacy	−0.77	0.37	0.03	−1.49 −0.06
Distance to the nearest healthcare centre	−0.38	0.15	0.02	−0.68 −0.07
***Need factors***				
Perceived health	0.22	0.17	0.19	−0.11 0.56
Disability	0.04	0.18	0.81	−0.32 0.41
Health district (Kaya reference)				
Ouargaye	−0.09	0.18	0.59	−0.44 0.25
Diebougou	1.31	0.20	0.00	0.09 1.70
Gourcy	1.75	0.28	0.01	1.20 2.31
_cons	0.81	0.42	0.06	−0.02 1.65

**Table 5 ijerph-17-06543-t005:** Regression model on service use (conditional upon reporting ill).

Variable	Regression Coefficient (β)	Std Error	*p*-Value	[95% CI]
***Predisposing factors***				
Age	−0.01	0.00	0.00	−0.02 −0.01
Sex	−0.31	0.15	0.04	−0.61 −0.01
Marital status	0.17	0.13	0.17	−0.07 0.42
Status in the household	0.42	0.13	0.00	0.16 0.68
Household size	0.08	0.03	0.02	0.01 0.14
***Enabling factors***				
Possession of user fee exemption card	−0.07	0.20	0.73	−0.45 0.32
Education	0.45	0.35	0.20	−0.24 1.14
Basic literacy	−0.25	0.1	0.42	−0.85 0.35
Distance to the nearest healthcare centre	0.00	0.13	0.97	−0.25 0.26
***Need factors***	
Perceived health	−0.56	0.18	0.00	−0.92 −0.203
Disability	−0.37	0.13	0.00	−0.63 −0.121
Health district (Kaya reference)				
Ouargaye	0.95	0.18	0.00	0.60 1.30
Diebougou	0.14	0.16	0.38	−0.17 0.45
Gourcy	0.10	0.18	0.58	−0.25 0.45
Time	−0.26	0.18	0.16	−0.62 0.10
_cons	1.12	0.34	0.00	0.45 1.79
/lnsig2u	−0.57	0.50		−1.54 0.41
sigma_u	0.75	0.19		0.46 1.23
rho	0.15	0.06		0.06 0.31
LR test of rho = 0: chibar2(01)	5.93			
Prob >= chibar2	0.01

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
