# Peer review of "Do Targeted User Fee Exemptions Reach the Ultra-Poor and Increase their Healthcare Utilisation? A Panel Study from Burkina Faso"

_ijerph, 2020, doi:10.3390/ijerph17186543_

Round 1

Reviewer 1 Report

The authors' work has centered on a study that uses an extensive sample of ultra-poor people (citizens of Burkina Faso), who were tracked over time to examine factors that led to the possession of user fees exemption card and the effects of this card possession on their utilization of healthcare services.

The first part of the document presents the environmental and application context of the proposed study in a very extensive way, also highlighting the ultra-poor selection and exemption mechanism and the reimbursement procedure within performance-based financing (PBF) program.

As a conclusion the study found that the utilization of healthcare services by the ultra-poor was not responsive to the introduction of targeted user fees exemption. To these conclusions the authors come through statistical analyzes reported in section 3 (Results).

Although the topics are interesting, it is quite evident that the scientific contribution is limited (only statistical and metric data extracted from a limited population of specific regions of the country in question are reported). Moreover, reading the paper is quite difficult, often we must refer to concepts described in previous sections in order not to lose the logical thread of the speech. In addition, the English form is to be improved heavily.

Some minore issues:

1) avoid very frequent repetitions in the text

2) standardize the writing of formulas, for example observe formula (1) and formula (2)

3) Do not use the first person plural (we used ... we examinated) but
structure the sentence like this:

- We examined the factors that led to the receipt of user
fee exemption cards ---> The factors that led to the receipt of user
fee exemption cards was examinated

- We used a panel data set of 1652 randomly selected individuals
identified as ultra-poor ---> A panel data set of 1652 randomly selected individuals identified as ultra-poor was used

4) Improve the quality of figure 1 (legends are not clearly visible)

Reviewer 2 Report

The Authors evaluated phenomenon of user fees exemption card possession on ultra-poor people utilization of healthcare services in Burkina Faso. This study is the first that uses an extensive sample of ultra-poor people, who were tracked over time to examine factors that led to the possession of user fees exemption card and the effects of this card possession on their utilization of healthcare services. This paper is a significant contribution to the scientific discussion about healthcare access and financial protection for the ultra-poor in Burkina Faso. It has a good scientific quality (methods, statistical tests, quality of discussion, and presentation of study limitations).

I propose to round the numbers to 3 (or 4) decimal places in tables: 3 and 4.

Reviewer 3 Report

Dear authors,

I would like to congratulate you for the work and amount of effort made to gather and write the current manuscript. I consider is highly interesting and the findings have relevance repercussions for the health public field.

Nevertheless, I would like to highlight some issues that need to be modified in the current manuscript.

  1. The affiliation of each author does not follow the guidelines of the journal. The authors need to delete the email 
  2. The abstract is 347 words and the limit set for the journal is 200 words, so it is necessary to reduce the current abstract by 147 words. 
  3. The introduction is no justified, and it should be rewritten. The current introduction section and context of study is a little bit messy. 
  4. The materials and method section has been described with section or parts that should be placed in the introduction or context. Also, the type of study is missing. The data and sampling are better written, but the data collection should include further information, such as, mean time to gather the data or how was the such data. The analysis could be placed after the variables and measurements, and the initial analysis should include the Kolmogorov-Smirnov test.
  5. The results section has really interesting data, but it should be further explained and highlight the significance of such findings. 
  6. The discussion section is so much better written and highlights the importance of the data. I really enjoy reading it, but there are some parts that I missed like the future research or the implications for other researchers, population or even the government.
  7. The conclusion could be further explained based on the relevance and the amount of data that the authors provided.
  8. In the reference section, I think that there are some articles that have been included more as webpages that articles, please check the reference section.

Reviewer 4 Report

General review:

This study is interesting and copes with the extremely important topic of using healthcare fee exemptions by the ultra-poor in Burkina Baso. However, it is very long and complicated, which greatly limits the enjoyment of reading, and if it were shorter and more concise, it would certainly bring a lot to this research area. The introduction contains information not directly related to this research. The material and methods section contains exact formulas that could be found in an appendix. The results are also written inconsistently. The work generally has great potential, however, it is written more in a book style. Therefore, I propose to shorten and synthesize this work and resubmit it again.

Detailed review:

Abstract

Correct abstract according to journal guidelines

Introduction

The introduction is too long. In my opinion, the introduction and context of the study should be combined and only the most important information should be included. Line 73-86 concerns fee exemptions on the health service utilization of the poor in other countries. These types of references should be analyzed in the discussion. In this study, the authors included 4 of the 8 Burkina Faso districts where we conducted a pilot program for fee exemptions. However, the study itself does not apply to these programs, so there is no need to describe them as precisely. Certainly, the paragraph on Reimbursement strategy is not necessary, because this program is not the target of the study. So far, the introduction reads like a book, each element is accurately described, however, it is not completely consistent with the purpose of the article. Therefore, I suggest a synthesis of the introduction. I also propose to clearly define the aims of the work and research questions, as you refer to them later.

Methods

Study design and conceptual approach is a wordy paragraph. It includes references to research questions that are missing and hypotheses that should be in the introduction. Conceptual approach should also be in the introduction.

Line 262 – I suggest referring to the significance level as a α ≤ 0.05.

In statistical analysis, I propose to put only a short description of the modeling and move the full descriptions of the models to the appendix.

Results

Please standardize the way of writing numbers. You write 1,652 once, and then 1652. Same with the tables. In the tables, also standardize the decimal places.

Minor issues:

Tables count is mixed, after table 1, there is table 1 again

Line 115- correct dot before the citation

Line 116 – put a dot at the end of the sentence

Line 157 – put a dot after a citation

Line 207 – remove a dot at the end

Line 256 - remove a dot at the end

Line 314 - remove a dot at the end

Line 371 - remove a dot at the end

Line 389 – remove a dot before p-value

Round 2

Reviewer 1 Report

The improvements made by the authors are sufficiently thorough.
In the first place, the length of the contribution was reduced,
and this made it smoother and easier. Furthermore, the entire document
has been structured in a coherent way with a scientific article
and definitely improved from the point of view of the grammatical form.
My final judgment is that it can be published.

Reviewer 3 Report

I would like to congratulate again the authors for the incredible work and the great revision made. The current manuscript has been modified to improve the reading and the understanding of the topic. After all the modifications, I have highly enjoy reading the new version.

Reviewer 4 Report

I feel that the authors did an excellent work in updating their review and now the new text is ready for publication.